# Generations and Life Worlds: The Case of Braga in Portugal

Eduardo Duque [1],* and José F. Durán Vázquez [2]

1   Faculty of Philosophy and Social Sciences, Portuguese Catholic University, 4710-362 Braga, Portugal
2   Department of Sociology, Political Science and Management and Philosophy, Universidade de Vigo, 36310 Vigo, Spain; joseduran@uvigo.es
*   Correspondence: eduardoduque@ucp.pt

**Abstract:** The objective of this research is to present the primary results of a study on generations and generational relationships in Braga, Portugal, specifically in the spheres of *family, school, work, leisure and consumption, and religion*. For this purpose, three generational cohorts were differentiated, belonging to those born in the decades *1940–1950*, *1965–1975*, and *1990–2000*. This work will focus on the first of these generations, *1940–1950*. To carry out this research, each generation was structured based on gender and social class, and qualitative (*Life Stories*) and quantitative (questionnaires) techniques were used. We started from the hypotheses suggested by previous research (blinded for review) and confirmed by the one being developed that it is possible to observe certain generational connections, despite the divisions related to gender and social class that exist within them (a division that gains greater visibility in the qualitative dimension), and that relationships between generations cannot be approached from theoretical models based on mere reproduction or opposition. Instead, we must grasp these relationships through more complex and dynamic processes, through which they will reconfigure and rework what is transmitted and what is received by each generation, resulting in a unique outcome that will be explored in this work. The religious dimension, which was previously a naturally assumed and lived tradition, ceased to be recognised as an identity determinant by the younger generations. As a result, the cultural heritage is no longer passed down as tradition. In this sense, there is a whole "integral ecology" (LS 159), in its broadest sense, recovering that is capable of cementing principles and values that unite generations and give meaning to life. To carry out this research, a theoretical and analytical framework has been established based on the following dimensions: *transmissions* (concerning the narratives and the people who appear as their referents); *temporalities* that articulate both the relationships within each generation and between generations; *spaces, structures, and rituals* that make up and give meaning to the transmissions; recognised *debts, obligations, and interdependencies*; *expectations and achievements*; and *tensions* between the *expected* and the *achieved*.

**Keywords:** generations; life worlds; family; school; work; leisure; religious; Portugal

## 1. Introduction

The subject of generations was analysed in the early twentieth century by the Spanish philosopher Ortega y Gasset ([1942] 2008) and the German philosopher and sociologist Karl Mannheim (1993). While it has been largely ignored since its conception, it regained momentum in the late 1980s (Corsten 1999), often due to categories created by various media rather than more or less relevant research.

The reason for the lack of attention that generational studies have received for so long may be related to the following circumstances. One is that, in the period following the Second World War, particularly from the 1950s, the Western world, especially North America and Europe, witnessed a generational conflict driven by the need for members of new generations to break away from their predecessors, to claim a new way of living and being in the world that was less sacrificial and also more independent and freer than that of their parents (Feuer 1968; Mendel 1975; Ricard 2001).

All this was also contributed to by the cultural climate created in Western societies in the 1960s, amid the upheavals brought about by the *May 1968 movements*, where youth became the main protagonist and the model for a cultural change aimed at constructing a new world free from the legacy of previous generations (Mead 1971, p. 109). Also contributing to this climate was a certain intellectual atmosphere, especially related to some sociological currents, such as *reproduction theories*, led by Bourdieu (Purhonen 2016; Le Goff 2002, pp. 356–57; Ferry and Rénaut 1985, pp. 199 ff.), Foucault's critical theory (Dubet 2006, p. 43; Habermas 1993, pp. 308–309), who stated that "had it not been for May '68, I would never have ended up doing what I am doing today" (cited in Pardo 2000, pp. 28–29), and also some *educational currents*, such as those linked to the *theories of deschooling* (Illich 1973). All these currents criticised, from different perspectives, various institutions, particularly familial and educational ones, for being a manifestation of a dominant legacy that prevented, by imposing its legacy in the form of different transmissions, the development of what was most genuine and authentic in the new generations.

In Southern European countries like Portugal and Spain, a favourable environment for youth was also generated, tasked with leading the change from dictatorship to democracy, with the will to leave behind their parents' world, identified with tradition and economic, political, social, and cultural backwardness that needed to be overcome (Leccardi and Feixa 2011). In this scenario, various sociological publications proliferated, which precisely had youth as the main protagonist (Instituto nacional de la juventud 1975; Instituto de Ciências Sociais 1987, 1988; Pais 1989, 1996, 1998; Nilsen 1998; Schmidt 1989, 1990; Martín Serrano 1994; De Miguel 2000; Guerreiro and Abrantes 2003).

The centrality that youth acquired in this social, intellectual, and political context favoured a perspective much more focused on the present and the future than on the relationship with the past and with previous generations, a sign of a conservatism that needed to be overcome, with youth being called upon to lead such an overcoming to advance towards a new and better world (Leccardi and Feixa 2011). The relationships with these generations were thus observed from the perspective of conflict, opposition, or mere overcoming, without contemplating, in most cases, other possibilities that would allow for an exploration of the complexity of such relationships (Mendel 1975).

In this context, where youth became the central leading figure, new concepts emerged that sought to label the new generations of young people, with names such as *Generation X*, the *Millennials, Generation Z*, and some others (Feixa 2014).

In what follows, we want to delve into the complexity behind the *concept of generation* and *generational relationships* based on the results of research we are conducting on three generations of people belonging to the geographical area of *Braga* in *Portugal* born in the periods 1940–1950, 1965–1975, and 1990–2000 in relation to *family, school, work, leisure and consumption*, and *religion*. This article focuses on the first of these generations, those born in the *1940–1950* decade. In this particular generation, *religion* has emerged as a fundamental dimension, because it articulates and gives meaning to the other spheres of life we analysed, as will be seen later. Indeed, it permeates their value system, giving it particular meaning, as well as almost all of their experiences and life situations. Religion also plays a role in the relationship that people establish with their environment, offering moral and ethical principles for the management of natural resources and care for the Earth. In this sense, "Laudato Si" (hereinafter, LS) highlights the concept of a "common home", stating "This sister now cries out to us because of the harm we have inflicted on her by our irresponsible use and abuse of the goods with which God has endowed her" (no. 2). This call resonates across all generations, serving as a reminder of the shared responsibility towards our planet.

In this way, the members of this generation incorporate eco-theological notions into their own way of living and understanding the Earth as a common home, a sacred space, cohabited by God and humans, which demands intergenerational respect and care.

Acting in this way, religion weaves bonds of interdependence between people and between these and their natural environment. This interdependence is reinforced by LS, which suggests that everything is closely interconnected, saying "the intimate relationship

between the poor and the fragility of the planet, the conviction that everything in the world is connected, the critique of new paradigms and forms of power derived from technology, the call to seek other ways of understanding the economy and progress, the value proper to each creature, the human meaning of ecology, the need for forthright and honest debate, the serious responsibility of international and local policy, the throwaway culture and the proposal of a new lifestyle" (no. 16).

And in this way, it also gives particular meaning to the obligations and sacrifices undertaken, both in relation to previous generations and to those that will follow. Religion also allows the tensions generated by different life trajectories to be faced, establishing bridges of meaning between the sphere of transcendence and that of immanence (Gil-Gimeno and Ibargüen 2023).

## 2. The Issue of Generations

Analysing generations involves first resolving the key question of whether there are specific circumstances that can create bonds between people born at the same time and place. For this fact alone, the mere *generational position* does not enable the creation of such generational bonds and, therefore, of a specific generational identity, although it may act as a substrate for these to emerge (Mannheim 1993).

However, for this to occur, there must be a series of common experiences and life situations from which similar attitudes and values, similar dispositions, and also a particular consciousness emanate (Mannheim 1993).

The question that arises in this context is how these experiences and these life situations, and the relationship that can be established from them with the past with what has been transmitted by previous generations in a specific social and cultural context, can favour the formation of certain generational bonds, creating a particular generational consciousness. As St Augustine said, the past and the future are hidden in their causes and promises; the present is all we have (St Augustine 1987) (Agustín 2010). This thought resonates with the need to consider how the past influences present generations in forming their own identity.

Analysing all of this involves first confronting the fact of *birth*, the appearance of new human beings in a world that precedes them (Arendt 2003, pp. 285 ff.) and which they must face in one way or another to build their own world amid the circumstances they have to live through. Without this essential event, there would be no generational change (Mannheim 1993, p. 212), nor would it be possible to revitalise the world, which would be growing old with nobody to renew it with new energies and new ideas (Arendt 2003, pp. 295–96). Each new generation thus carries a vital energy that those who are older in their world lack (Mannheim 1993).

However, the emergence of new generations also entails the encounter between what was transmitted by previous generations—that is, what is inherited—and what is experienced and experienced by oneself. Without this encounter, the world would be in perpetual renewal, without any perspective of past or future, or it would die of old age, as new people would not enter it, who must face the inheritance received to build their own world (Arendt 2003, pp. 295–96, 1998, pp. 265–66). This meeting occurs from the present, because, as Saint Augustine said, the past and the future are hidden in their causes and promises; the present is all we have. "There are three times—he writes: present of past events, present of presents and present of futures. In fact, these three do somehow exist in the soul, and I do not see them anywhere else: present memory of past events, present contemplation of the present and present expectation of the future" (Agustín 2010, book XI: p. 567). What memory does each generation have of past events that makes them contemplate present events in a certain way and have a certain perspective of the future?

Generational relations thus involve both memory and forgetting (Mannheim 1993, p. 213), both looking back and looking beyond (Joas 2021, p. 189)—forward. All of which is conditioned by the fact that the older ones, who carry the weight of an entire past experience with their own struggles and contradictions, tend to see the new under the prism of that

experience, while the younger ones lean towards the opposite, to observe and interpret all the previous inheritance under the gaze of the new, of their particular experiences and life situations, as well as of their own conflicts and contradictions. And it is also conditioned by the fact that what is experienced for the first time during youth is recorded in such a way that it acts as a substrate that reconfigures subsequent experiences and life situations and also those transmitted by previous generations (Mannheim 1993, pp. 216–217).

The encounter between the inherited, the experienced, and the lived can thus result in opposition and denial or in problematisation and re-elaboration of the received inheritance and of one's own previous experiences and life situations in the reproduction of a previous legacy or simply in the disconnection from the previous tradition based on a will, more or less explicit or implicit, to assert oneself in the present.

Amidst this complexity, the issue of generations must be analysed, which is closely linked, as we have seen, to that of the *continuity* and *reproduction of societies* (Arendt 2003, p. 296). This complexity is even greater, since, often, generational divisions are crossed by other social divisions, such as those of classes. And all of this becomes even more complicated when class becomes an all-encompassing concept as a total explanatory fact that conditions and comprises all other social relations (Purhonen 2016; Alexander 2000). From this perspective, they appear as realities constructed in the context of different competitive struggles to maintain different social positions. "My point", wrote Bourdieu from this perspective, "is simply that youth and age are not self-evident data but are socially constructed, in the struggle between the young and the old" (Bourdieu 1993, p. 95).

With the reification of the objective reality of classes, converted into the *factum* par excellence, any other concept is subjected to criticism, such as that of generation, which claims the same explanatory capacity, and also any other consciousness, more or less manifest, that does not refer to said reality.

On other occasions, what appears oversized is the character of this consciousness, of generational consciousness, especially when it is attributed a political meaning (Purhonen 2016; Aboim and Vasconcelos 2014) associated with the idea of knowing oneself to be part of a world, particular and proper, different from the previous one and with a renewing and transforming will.

However, this consciousness is not always manifest. Indeed, people who are born at the same time and place, and under certain circumstances, with more or less similar experiences, attitudes, and values, do not necessarily have to have a full generational consciousness (Aboim and Vasconcelos 2014, p. 176). This can be linked to a certain practical sense, to a *Habitus*, to practices and representations that, as Bourdieu said, "may be objectively adapted to their goal, without assuming the conscious purpose of certain ends or the express mastery of the operations to achieve them" (Bourdieu 2007, p. 86). In this case, certain social and emotional dispositions are created in the subjects that lead them to act, to think, and to feel in a certain way.

On other occasions, however, this consciousness can be more rational and reflective if the subjects receive what is transmitted with a certain attitude of *estrangement* (Mannheim 1993, p. 214).

Taking into account everything said so far, we consider that, under certain common social and historical circumstances, a way of being and understanding the world that identifies—and with which certain social groups identify—can emerge, although, on many occasions, this identification is not deliberate and conscious and is not accompanied by a full sense of generational belonging. From this perspective, we can understand the *concept of generation* as a constructed reality, often in a more practical than rational and reflective way, in relation to certain life experiences and what is transmitted by previous generations in a certain social and cultural context.

Analysing generations and generational relations therefore requires close contact between theory and the investigated reality, so that the theory illuminates new paths to approach that reality, and this provides information not contemplated by the theory but capable of being analysed and interpreted in light of it. From this perspective, we have

analysed the generation of those born in 1940–1950 in the *region of Braga* in *Portugal*. Next, we will show the results we have obtained.

### 3. Materials and Methods

The members of the generation under our investigation are part of a social and cultural context in which the transition is made from the traditional world to that of production capitalism and, from this, to consumer capitalism, as we have shown in previous works (blinded for review). The generation in question in this article (1940–50) belongs to the first of these stages.

Members of said generation were structured based on *gender* and *social class*. For this, we differentiated an *origin class*, low and lower-middle, given the scant existing middle class at that time in Portugal prior to the country's modernisation process, and a *destination class*, lower-middle and middle, when this modernisation process had already started, with the subsequent increase in social mobility. Social class was determined based on the *type of work* and the *level of education*.

To collect information, we mainly used qualitative techniques (*Life Stories* and *Focus Group*)—specifically, we conducted 8 *Life Stories* (4 to women and 4 to men, belonging to the origin and destination classes previously indicated) and 1 *Focus Group*. We also designed a survey to contrast the results obtained in qualitative interviews and check if the macro data provided any information not contained in the qualitative interviews.

*Life Stories* have proven to be a fundamental instrument for collecting information from our informants for two fundamental reasons. The first is that, being people with a long biographical journey (over 70 years), their lives are understood from the perspective of this life trajectory from which the past, the present, and the future take on meaning (*see Bertaux, life stories*). This trajectory, formed over time, reveals itself, with all its contradictions, when narrated. This is how the reference frames and the horizons of meaning that have guided it also come to light, as well as the social and moral frameworks to which they are linked, from which one understands and judges one's own life and those of others (Taylor 1996, 42 ff.).

The second reason that has led us to use *Life Stories* in our research is that, since the first impressions of childhood and youth are those that leave a more special imprint on people's lives, from them, subsequent experiences and life situations are shaped (Mannheim 1993, p. 214); a life story is the best way for this lived and experienced reality to come to light (Bertaux 2005).

The choice for the *questionnaire* as a data collection instrument allowed us to formulate a series of questions related both to the objectives of the study and to the socio-professional situation of the respondents.

In constructing the questionnaire, before formulating the questions, we sought to identify a set of theoretical dimensions, and for each of these, we established a set of indicators aligned with the objectives of the study. In this way, we defined seven dimensions: namely, with the first dimension, we intended to analyse the individuals' relationship with work, including the perception of job satisfaction, the balance between work and personal life, and the influence of work on the identity and well-being of individuals. With the second dimension, we sought to study the structure and family interactions, including socialisation practices and the values passed down through generations. The third dimension aimed to explore how school and education influence individual development, attitudes, values, and skills. The fourth sought to study the role and influence of religion in society and in the lives of individuals. The fifth dimension aimed to study leisure and consumption, including shopping habits and attitudes towards leisure. The sixth dimension focused on the analysis of questions about the level of education of the parents to study social mobility and the importance of education in this mobility. The seventh and final dimension explored sociodemographic issues so that, when cross-referenced with the other questions on the questionnaire, a more integrated approach was achieved on how different sociocultural and institutional factors shape the work experience of young people.

For the interpretation of the interviews and the analysis of the questionnaire responses, we resorted to *content analysis*, a method that allowed us to identify recurring themes and patterns and categorise and code the data, transforming the responses into analytical categories, which provided us with a more solid structure for our interpretation and subsequent conclusions.

The option to combine qualitative approaches with quantitative ones, commonly referred to as *data triangulation*, strengthens the robustness and reliability of the results. This diverse set of methods allows us to approach the phenomenon studied from multiple perspectives, minimising the limitations inherent in each individual approach and providing a more complete and accurate analysis of the study in question.

As for the *sample* for the questionnaire application, as previously mentioned, people born in the periods 1940–1950, 1965–1975, and 1990–2000 were surveyed, both in person and online, through the convenience probabilistic method. In this text, as we have also mentioned, only the data relating to those born in the decade 1940–1950 will be analysed.

In the following, we will show the results obtained in our research based on the theoretical and analytical framework previously described.

## 4. Results

The people of the generation that we investigated, born in the 1940s–1950s, were raised and socialised in farming families, more or less humble and integrated into traditional communities. This event will mark these people, as will be seen later, in various aspects, and particularly in the relationships that they maintain with the land and with the natural environment. These families were generally large, with several children, in which the father was the *breadwinner*, who maintained a disciplinary and authoritarian position, not being close or affectionate. "My father was more rigorous... my mother was not so rigorous", said one of the people interviewed in *Life Stories* (hereinafter, LF) (LF: *woman with six children. Unfinished primary studies. Suitcase factory employee. Widow. very poor social origin*), and another commented "my father was rigid, he was not a close person" (LF: *woman with two children, retired primary school teacher, social origin, small agricultural owners*). "My father especially", said another of our informants, "he set a time for me to enter the house, if he went beyond that limit, he would hit me. He was strict, I had to follow the orders he transmitted" (LF: *typographer, primary education, retired with three children, poor social origin*).

The mother, on the contrary, appears in our interviewees' stories as a much closer and affectionate figure, who assumed the role of caregiver, educator, and manager of the home: "My mother", a woman told us, "was less rigorous... she was more affectionate" (LF: *woman with two children. Primary education, typist secretary from a retired driving school. Humble social origin*). "My mother", commented a man, "got up later than my father, she made breakfast. If we (the children) had not yet gotten up, she would bring us breakfast to the room and sit next to the bed to talk to us (...). She was very loving, she was a mother" (LF: *man with two daughters, technical engineer, retired secondary school teacher. Social origin small agricultural landowners with some day laborers in their care*). "My mother was the one who gave us education", said another of the people interviewed (LF: *woman with two children, retired primary school teacher, social origin of small farm owners*). The mothers of the people we interviewed in our *Life Stories*, belonging to the traditional peasant world and of more or less humble social origins, not only performed the role of caregivers and educators of their children and administrators of the home but also worked in the fields, either on their small property, in a self-consumption economy, or working for others, in the case of the poorest families. "Mi madre," commented uno de nuestros entrevistados, "se levantaba más tarde que mi padre, hacía el desayuno. Si nosotros (los hijos) aún no nos habíamos levantado, nos llevaba el desayuno al cuarto y se sentaba al lado de la cama a hablar con nosotros (...) She would go to the farm in the middle of the morning to pick vegetables for the meal" (LF: *man with two daughters, technical engineer; retired secondary school teacher. Social origin small agricultural landowners with some day laborers in their care*). "My mother had her land, her field," said another of our informants, "she raised her poultry, baked

bread, worked in the garden, raised the animals... she cooked, she washed... (LF: *retired small businessman. Four children. Primary education. Poor social origin*). For this reason, by maintaining closer and more affectionate contact with her children, the figure of the mother remained more alive in the memories of the people we interviewed than that of the father. "I'm going to be frank", said one of them, "the most important person in my life was my mother" (LF: *retired small businessman. Four children, primary education. Poor social origin*). The mother was also in charge of transmitting and instilling both religious values and practice in her children. "There was a lot of religious practice in my house. My mother went to mass on Sundays and took us with her", said one of our interviewees (LF: *woman with two children. Primary education, typist secretary from a retired driving school. Humble origin*). "Since I was a child, my parents, especially my mother, took us to church", said another (LF: *man, retired primary school teacher with two children*).

Our informants, as we said before, grew up in dense families, generally with several siblings, in which they internalised from a very young age values, attitudes, and forms of behaviour that were not transmitted through persuasive, reasoned, and argued speeches, which invited reflection to those who listened to them, but through other means, verbal and non-verbal, such as actions and gestures, images, narrations and descriptions, repeated religious prayers, examples, and expressions. All of which made sense and was assumed and internalised in the dense spaces of interactions in daily life. "Everything my parents transmitted to me was more by example than by word. The example of life conduct", said one of the people interviewed (LF: *man with two daughters, technical engineer; retired secondary school teacher. Social origin of small farm owners*). "I learned from the experience of others", said another, adding with special emphasis, "that they told me (...) Because that's how I say, people learn, people observe (...) My mind was forming with what I was hearing" (LF: retired small businessman. Four children, primary education. Poor social origin). All these transmissions made sense in the different social settings in which daily life took place. "My mother", commented one of the people interviewed, "was at home cooking and she asked me what colour the priest's cape was, and I had to tell her" (LF: *retired small businessman. Four children. Primary education. Poor social origin*). Having thus received these transmissions, this was how several of the people we interviewed transmitted their life teachings: "I am going to tell you a story that I tell my grandchildren about giving everything to the children", said one of them to introduce his story (LF: *man, retired primary school teacher with two children*). "I tell an episode that, frankly, made me feel bad", said another (LF: *man, three children. Retired typographer. Primary studies. Poor social origin*). And another commented, "let's take specific cases, how bread was made, for example; how the beans and peas were planted in the garden, how the gardening was done, the irrigation was done" (LF: *retired small businessman. Four children. Primary education. Poor social origin*).

Living and interacting with other people within hierarchical and positional structures was how they were transmitted and instilled—through images, actions, gestures, examples, narratives, and expressions first in the family and then in other spheres of life—values, attitudes, and forms of behaviour that signalled to those who received them their position within said structures. These transmissions were therefore related to what the sociologist *Basil Bernstein* called "*restricted codes*"—that is, inclusive forms of communication, closely linked to the interaction contexts of everyday life, in which *how* it is said matters as much or more than *what* it says—and that they are fundamentally narrative and descriptive rather than abstract and analytical ([Bernstein 1989](#), I, 85 ff.). People socialised in this type of codes, therefore, end up conceiving themselves as part of the world, of their world, and not as individuals facing the world ([Bellah 2017](#), p. 72).

This is how the people we interviewed received and internalised a series of values that appeared again and again in their life stories. These values were those of *honesty* and *trustworthiness*, *respect*, *solidarity*, *justice*, *responsibility*, and *work* and, also, that of *social mobility*, linked above all to the work and educational trajectories of peasant families who were a little more comfortable with some land properties who could have certain social aspirations. Responsibility and being trustworthy and honest were, precisely, the most

important qualities most people of this generation that we surveyed said that their families transmitted to them (25.6% and 17.8%, respectively, of those surveyed pointed this out. See survey, graph 1). These two qualities were also among the three most important that the majority of people surveyed indicated that they would like to transmit to their children, without differences related to gender being appreciated in both cases (*see survey, graph 2*) or at the level of education; the most important qualities that their families transmitted to them: $\chi^2$ (28) = 39,42, ns; the most important qualities to transmit to children: $\chi^2$ (26) = 14,95, ns).

If we exclude this last value, the other values were impregnated with a clearly *religious* meaning linked to the beliefs and practices of the Catholic religion. "The values that I have transmitted to my children, associated with religious practice, are there", "commented one of our informants, "the values of honesty, respect, that no one is more than anyone else, although recognizing one's own merit" (LF: *woman, retired primary school teacher, two children. Social origin small agricultural owners*). Some of these values appeared among the three most important that the majority of people surveyed from this generation said they received through religion, specifically the value of generosity (28.6% of people surveyed), being honest and trustworthy (14.3%), and having your word and trusting others (11.9%) (*see survey, graph 5*). Without significant differences being seen regarding the gender and level of education of the people surveyed: gender ($\chi^2$ (8) = 7,87, ns); education ($\chi^2$ (16) = 18,32, ns).

Religious values, which, as noted before, give meaning to almost all other values, were received by the people we interviewed, and they also wanted to transmit them to their children, not only through words but also through repeated and continued practice. It was a religion of both believing and doing, typical of the popular religiosity (Taylor 2014, vol. I, p. 112), in which beliefs and practices were closely linked: "There was religious practice in my house", said one of the women interviewed (LF: *woman with two children. Primary education, retired typist secretary from a driving school. Humble social origin*). "Being Catholic or not being Catholic meant nothing to me", said one of the people interviewed. What meant to me were the doctrines that were practiced at that time. At my mother's house it was praying the rosary. Before, it was not like today, there was a church that I went to, because my mother forced me" (LF: *retired small businessman. Four children, primary studies. Poor social origin*). "At home we prayed almost every day at night", said another person" (FG: *Male. Humble social origin. Primary education. Three children. Worked for 50 years in a textile company as a laboratory manager. Retired 10 years ago*). And another of the people interviewed commented, "since I was a child, my parents, especially my mother, took us to pray the rosary on Sunday afternoon... My mother always took us to the church to participate in the celebrations, not only in mass, there was also the Christmas novena at that time" (LF: *Male, retired primary school teacher. Humble social origin*). "My father", said another of our informants, "taught us to pray and follow our Catholic religious life. He taught us and gave us a very good education. He taught us to do good, to go to the Catholic church. It was the good thing that he left us, and so did my mother" (LF: *woman with six children. Primary education unfinished. Suitcase factory employee. Widow. Very poor social origin*).

This close relationship between belief and religious practice, typical of the religiosity of the people we interviewed, was what was later missing in the children. "I also instilled in my children to go to mass", said one of our informants, "but they are already married, and I know that they don't always go... They say that they have other occupations and that is why they don't always go to mass on Sundays." (...). They are Catholic, only they don't go to mass as I would like them to, because they say they don't have time" (LF: *woman with two children. Primary education, retired typing secretary from a driving school. Humble social origin*). "It especially hurts me that my children do not follow the religious values that I transmitted to them", said another of the people interviewed, "although at first they went to mass with me, later they began to make excuses" (LF: *woman, retired primary school teacher, two children. Social origin small agricultural owners*). "When they were younger", said another, also referring to her children, "they started going with me, but then they started to stray a little, but they are Catholics (LF: *retired typographer, primary education. Three children, poor social origin*). And another of our informants commented that, although his daughters

continue with the religious values that were transmitted to them, "they doubt more", he said, "than I doubted, mainly in religious practice" (LF: *man with two daughters, technical engineer; retired secondary school teacher. Social origin small agricultural owners with some day laborers in their charge*). As can be seen, while religion was a lived and experienced reality for the parents, for the children, it was not lived or experienced in the same way, especially as they grew up. For the former, it was a plausible and taken for granted reality based on which, they interpreted the world, while, for the latter, it was neither plausible nor taken for granted (Berger 1977, 65 ff.), since they wondered about religion from the possibility of disbelief (Taylor 2015, vol. II, pp. 47 ff.). The transmitted belief thus gradually languished, as it was not revitalised through daily practice, because the practice involves introduction into a ritual and symbolic world without which the belief loses its previous meaning, as it can no longer be understood by referring to that practice (Douglas 1988, pp. 64 ff.). Along these lines, as belief is separated from practice and is gradually deinstitutionalised (Beck 2009, 94 ff), it ceases to be a fundamental constitutive part of one's own identity, losing the capacity to reconfigure subsequent experiences (Mannheim 1993, pp. 214–15). Hence, communication between each other, parents and children, becomes more difficult, increasingly distanced in their way of understanding religion and, therefore, also the world.

The religious transmissions that, as we have seen, the members of this generation received at first in the family environment through a close relationship between belief and practice continued later in other areas, such as the *parochial*. "There I discovered", said one of the people interviewed, "the value of living in society and sharing common values. Those meetings marked me so much that today I continue to belong to those parish groups" (FG: *woman, two children, retired school janitor, with primary education. Very poor social origin*). People who studied in seminaries, a means of social promotion for the children of humble families, continued their religious socialisation in these institutions. "Later", commented one of these people, "I entered the seminary, and naturally I followed the religion, and I maintain it to this day" (LF: *Male, retired primary school teacher. Humble social origin*). In other cases, said socialisation took place in religious institutions such as *Ação Católica*. "*Ação Católica*", commented one of our informants, "gave me a general culture (LF: *woman with primary education. Two children. School janitor Retired. Very poor social origin*). School institutions, many of them linked to the Church, were also a means of religious transmission for members of this generation. In one of these religious schools, one of our informants told us that she "prayed every day, especially in the month of Mary" (LF: *woman with two children, retired primary school teacher, social origin of small agricultural owners*).

People who have been thus socialised in the religious universe understand and judge the world, as will later be seen, through religious beliefs and values, because religion is part of their individual and social conscience (Berger 1977, p. 104). But religion also provides faith and confidence, reducing the uncertainties and tensions that life can cause, a faith in the existence of someone superior to whom one wishes to bond, seeking protection, security, and trust. "Religion", said one of our informants, "unites us. Faith has to have someone, I have to have someone who is superior, who I feel is superior (...). I have someone? Yes. I have someone to ask for help (...) Because what is faith? Why do I want to be religious? Because that's where I'm going to seek for my faith. And then the following happens. How I think; I am aware of any problem, but I have faith that God will help me (LF: *retired small businessman. Four children, primary education. Poor social origin*). From this point of view, religious faith involves the externalisation of individual conscience. In effect, the individual needs to project himself onto someone external to himself to find himself, to find a source of meaning that transcends his own subjectivity. This appeal to someone or something that transcends the sphere of the immanent is one of the main characteristics of religious beliefs (Taylor 2014, vol. I, p. 47). Although this appeal to transcendence has been interpreted as a source of alienation (Marx 2001), it can nevertheless act in another direction, providing a meaning to existence, both the happiest and joyful and the saddest and most miserable (Berger 1977, 104 ff.). If, even so, it can be said that religion alienates, it is not because it interferes with a supposed original and natural individuality subject only to its own

conscience, corrupting it to the extent that it transcends it (Marx and Engel 1994, p. 40). If religion alienates, it is, on the contrary, because it constructs worlds of meaning in which human beings project themselves fleeing from chaos and anomie (Berger 1977, 114 ff.).

Religion acts for the people we interviewed as a source of transcendence, allowing them to find meaning, stability, and order in their lives, permeating the main values that articulate the different spheres of their existence. To do this, individuals need to project themselves beyond themselves, into a higher reality that transcends them. The different values that they repeated over and over again—*honesty* and *trustworthiness*, *respect*, *solidarity*, *justice*, *responsibility*, *work*, and *savings*—imbued with a religious sense represent so many spheres of that transcendence.

Raised and educated in families and in peasant communities, the land is, for these people, one of the dimensions of this transcendence, because it provides meaning, order, and stability to the world. Therefore, this care must be respectful of the natural environment from which one benefits and which also benefits us. The relationship with the land thus incorporates many of the values mentioned above: *honesty* and *trustworthiness*, *respect*, *solidarity*, *justice*, and *responsibility*. This idea of caring for the earth so that it produces its fruits through a relationship of interdependence between human beings and their natural environment is presided over by the values of solidarity, honesty, responsibility, and justice, and it was transmitted to our interviewees by the peasant families in which they grew up. "My mother", said one of these people, "transmitted to me the value of respecting and caring for the earth and nature as well (…) My mother had her land, her field, she raised her birds, her rabbits (…). She baked the bread, took care of the garden, raised the animals, killed them…" (LF: *retired small businessman. Four children, primary education. Poor social origin*). These values were what our previous interviewee said he put into practice throughout his life. "For about 20 or 30 years I dedicated myself to a small garden on my property, I even went overboard. And that I ended up verifying after a few years (…). I prepare the land and put the seeds, I put the plants. I want them to grow. I don't want them to lack food or water. They begin to grow (…). I begin to think, my interior and my mind ended up being more humanised. I treat that with affection. That is humanism too (…) And then the fruit, the distribution of the fruit, the pleasure it gives" (LF: *retired small businessman. Four children, primary education. Poor social origin*).

However, this relationship of exchange between human beings and their natural environment was far from being, as we said before, egalitarian, since it implied the recognition of a certain natural order that is offered to human beings but that they have not created. Hence, the relationship of these people with the land is dominated by a certain attitude of gratitude for what they have received but do not fully deserve. This attitude is reflected in the following testimony, in which one of the people interviewed, raised in a peasant family of small landowners, told of the relationship that his mother had with the land. "My mother", he said, "would go to the garden mid-morning to pick vegetables for lunch, and when she returned, she almost always said, thank God, look what the garden has given us." (LF: *man with two daughters, technical engineer; retired secondary school teacher. Social origin small agricultural landowners with some day laborers in their care*). This attitude towards the land, which reflects the values of solidarity and gratitude, values that the people interviewed internalised in their peasant families of origin, was also what they themselves expressed in their respective life stories. "I helped with the work in the fields until I went to university", said one of these people, "and I also liked to help the neighbours, for example, picking potatoes. I always liked, and I still like, farm work, I find great satisfaction in it, because it reminds me that nothing can be achieved without the help of God and other people." (LF: *woman with two children, retired primary school teacher, social origin of small farm owners*).

When, on the contrary, this relationship of interdependence with the natural environment is broken, it is irreversibly degraded. "We are", commented one of our informants, "breaking the balance with nature, with the land, we abuse it, and therefore also those who have offered it to us for our own benefit… We are doing this and no one is going to stop it."

(LF: *man with two daughters, technical engineer; retired secondary school teacher. Social origin small agricultural landowners with some day laborers in their care*).

This way of understanding the relationship with the natural environment and, in particular, with the land as a relationship of unequal interdependence between nature, invested with a certain sacred character, and the human being, grateful for what it offers, is typical of pre-industrial societies. Thus, for example, in the ancient Greek world, agricultural occupations were considered to involve the relationship of the human being with a nature that had sacred attributes and that was subject to divine will, a nature on which the human being depended to ensure his subsistence and which he therefore had to face with the attitude of the warrior (Vernant and Vidal-Naquet 1988). This way of understanding the relationship with the land was very similar to that of Ancient Rome. Cicero therefore included agriculture among the noblest occupations in societies in which work had the worst consideration (Arendt 1998; Vernant 1985). "But among all the occupations through which something is acquired", he wrote, "the best, the most abundant, the most delicious and typical of good, is agriculture" (Cicero, First Book, chap. XLII, p. 80) (Cicerón 1995, First Book, chap. XLII, p 80). This way of understanding the relationship with the earth, as something that the Divine Creator offers to human beings so that they can work for their own benefit, was also typical of the medieval world, although interpreted in Christian terms (Le Goff 1983). And we could still find it much later in thinkers like Thomas Hobbes when he wrote "As for the abundance of materials, it is limited by nature to those goods which God gives us, either freely by making them spring from the earth and the sea–which are the two breasts of our common mother –, either in exchange for work (. . .). So, abundance will depend, after having been given to us by the favour of God, on the work and industry of men" (Hobbes 2004, p. 217).

Modern Western thought breaks with this way of thinking with respect to the earth and the natural environment when it no longer conceives this reality as something that the Divine Creator offers to human beings for their sustenance so that he can only intervene in it by bringing to fruition with his effort what was already potentially created. That is why his attitude towards the natural environment is one of admiration and gratitude. This rupture is clearly observable in the thoughts of John Locke: "I think", he wrote, "that it would be a very modest estimate to say that, of the products of the earth that are useful to man, nine-tenths are the result of work. Well, if we estimate things precisely as they come to us for our use. . .what they strictly owe to nature and what they owe to our work, we realize that in most of them ninety-nine percent must be attributed to our efforts" (Locke 2006, pp. 67–69). From this perspective, admiration and gratitude are no longer directed toward nature but toward human work, which, throughout modernity, will become the source of a sacredness and transcendence that previously resided in nature. Human work as a source of productivity will end up being one of the main signs of progress, identified with its capacity to transform nature, not so much for the benefit of the well-being of humanity but, rather, for the industrial system it serves (Bury 2009, 224 ffs.). And this progress, as is known, knows no limits.

The people we interviewed were much closer, on the contrary, as has been shown, to the conception of nature and the natural environment linked to the premodern mentality than to the latter that began to develop with the thoughts of John Locke. From this perspective, they find in the relationship with their natural environment, through the cycle of repetitions that it imposes, a source of stability and order that makes it possible to face the disorders that ordinary life produces. In this sense, nature, imbued with a sacred character, acts as a transcendent dimension that brings order where chaos would otherwise reign. The perspectives of the people we interviewed, aligned with a mindset that considers nature and the natural environment not as resources to be exploited but as a sacred and intricate extension of human life, is highly resonant with the teachings of the Encyclical Letter Laudato Si that highlights the interconnectedness of all living beings and our collective responsibility to the Earth. As Pope Francis observed, "everything is interrelated and



authentic care for our own lives and our relationships with nature is inseparable from fraternity, justice and fidelity to others" (no. 70).

According to Laudato Si, it is not only a question of *environmental ethics* but also of *social ethics*, as one cannot think of establishing an ethics of fraternity and friendly dialogue with the environment without social love. As LS no. 231 stated, "Social love is the key to authentic development: 'In order to make society more human, more worthy of the human person, love in social life—political, economic and cultural—must be given renewed value, becoming the constant and highest norm for all activity'. In this framework, along with the importance of little everyday gestures, social love moves us to devise larger strategies to halt environmental degradation and to encourage a *culture of care* which permeates all of society." This reality resonates with the feelings of our interviewees, for whom caring for the environment is not an isolated issue but an integral part of a balanced and ethical life.

Furthermore, the sacred conception of nature, expressed by our interviewees, is parallel to what Pope Francis in the LS described as "the value proper to each creature" (no. 16). Nature, in this sense, is not just a backdrop for human activity but a manifestation of divinity that deserves all our respect and care.

In this context, Pope Francis' words in Laudato Si were nothing more than a call to action and reflection for all humanity, suggesting that "integral ecology is also made up of simple daily gestures which break with the logic of violence, exploitation and selfishness" (no. 230). "The challenge now is how to translate this deep understanding and these sensitivities into policies and actions that protect the environment and promote a true "culture of care" (no. 231).

In this sense, the people we interviewed not only understand the relationship with their natural environment through their beliefs and religious values, but they also interpret in this way the relationship they establish with the other areas that articulate their existence. In all of them, religion intervenes in the same way, giving a deep meaning to the values they share, those of *honesty* and *trustworthiness*, *respect*, *solidarity*, *justice*, and *responsibility*.

The dense religious universe in which the people we interviewed were socialised, linked to a series of practices and values such as respect, honesty and trustworthiness, solidarity, justice, and responsibility, grant confidence and meaning to the relationships they have established with the nature they inhabit and with other people, even when their lives have been full of uncertainties.

All of this is related to a sense of time that is also stable and organised, such as the superior time of transcendence (Taylor 2014, vol. I, p. 109) or that of the repetition of the agricultural cycle or that other linked to the tradition of the values transmitted by previous generations or that of a long and continuous working life, which stability is also especially valued (*see survey, graph 6*). In this way, they can create islands of security in the middle of the ocean of insecurity in which they often have to live (Arendt 1999, p. 106), transferring the future to the present and relating the present to the past (Adam and Groves 2007, p. 47). This is part of the trust they intend to communicate to the following generations. Thus, 51% of the people we surveyed indicated, when asked about the phrase with which they most identify, the one referring to "my experiences, and what I learned and was taught in the past, help me live the present and trust in the future" (*see survey, graph 7*).

Religion also gives deep meaning to *work*. The people we interviewed understand work from a double perspective, both positive and negative. From the negative point of view, work was understood as a destiny that human beings had to carry in this earthly life to provide for themselves. He who does not work should not eat", said Saint Paul in the Gospel (Second Letter to the Thessalonians, 3). "It was that culture", said one of the people interviewed, "that we have to work, that we must work. In this world we have to work. He who doesn't work doesn't eat. It was this idea. We have to work. It was this" (LF: *woman, two children, retired school janitor, primary education. Very poor social origin*). However, if the hardships and sacrifices that the work required were more bearable, it was because this activity was associated with the values mentioned above, impregnated with a religious sense. From this perspective, work is a moral duty, without which fulfilment one

cannot become an honoured and honest, dignified, and respectable person. "And this", said one of the people we interviewed, "was what I taught my children. Do everything as God wanted, work, be honest, and try not to steal from the bosses (. . .). This was the most important thing my parents taught me, to be hard-working, serious, honest, and work and not be watching. . . and help others. . . it was what I taught my children, and it was what I did in my work" (LF: *woman with six children. Primary studies unfinished. Retired suitcase factory employee. Very poor social origin*). "All my work", said another person, "with a lot of suffering at times—I always continued to see it as a form of personal fulfilment, because if I don't work, for me it is a very great frustration because I don't do anything; even if I wasn't going to create anything, I was at least contributing to something" (LF: *female with primary education. Two children. School janitor. Retired. Very poor social origin*). One of the aspects of this appreciation of work, with a clear religious influence, is the boundness to social utility. "Later on", one of our informants told us, "I got to know *Ação Católica*. I started in pre-adolescence. . .there I began to develop, to have knowledge of various things, and to understand that, in fact, we human beings have value, and that work is an important thing for the development of society (. . .) Today I understand that our work has to be seen as help for social development. All jobs have to be seen from this point of view. I try to pass this on to my children; when they complain about their job, I tell them, try to think about what you contribute to the development of society, it may not be the best job, but it is the one you can help with" (LF: *female primary education. Two children. Retired School janitor. Very humble social origin*).

This aspect of work, linked to the values of honesty and respect, also appeared as one of the most prominent values in the survey we carried out (*see survey, graph 6*).

This sense of work, associated with a moral and social duty with a clear religious background, made the tensions derived from the need and obligation to work, with the sacrifices and hardships that all this entailed, not only much more bearable but also acquired greater meaning. One could thus feel more or less satisfied, being considered a worthy and respectable person. "My father told me", commented another of the people we interviewed, "that when the boss passed by, not to lower my head, to look him in the face, this way you show that you are doing your duty; and I did it like that. . . I always thought that, if I was responsible in what I did, I didn't have to lower my head" (LF: *woman primary education. Two children. Retired school janitor. Very poor social origin*).

Work, understood in this way as linked to some source of transcendence and permeated with a religious sense that elevated it to a social and moral duty, mitigated the tensions generated by work life, full of hardships and sacrifices. These sacrifices could also be compensated by the expectation, fully secular and linked to immanent ordinary life, of social mobility. But this expectation was, for the people of the humblest social status that we interviewed, more of a wish and a distant hope than a reality. Therefore, it could not nourish the meaning of a working life, which would otherwise fall into anomy (Durkheim 1995). "When I stopped studying, what was going through my head", one of these people told us, "many things, I wanted to be equal to someone, but I didn't have any ability or knowledge (. . .) what I had to do was take advantage of the job that arose to earn some money for everyday life, and that's how it was" (LF: *man, three children. Retired typographer. Primary studies. Poor social origin*). If, for these people, their work life had any meaning, it was, above all, in relation to the social and moral duties assumed, to which we have previously referred. Those other people we interviewed, who prospered more socially, experiencing processes of social mobility either from a very humble position or from one that was a little more comfortable due to being children of small agricultural owners, give greater meaning to their working life in relation to these processes. This sense, far from lowering that other one linked to the fulfilment of the moral and social duty associated with work, complements it. Thus, the same person who said that "through my work I managed to evolve, but for that I had to dedicate myself a lot to work" (LF: *retired small businessman. Four children. Primary education. Poor social origin*) also told us "In my house, unfortunately,

we had hardly anything… there were values, respect" (LF: *retired small businessman. Four children. Primary education. Poor social origin*).

Work, understood in this way, was not a matter of choice but rather an assumed social destiny, and as such, the free time that can be left to devote oneself to other occupations in life was barely valued (*see survey, graph 6*).

This sense of existence provided by religion, linked to the belief in some transcendence that also gave meaning to the fulfilment of a series of social and moral duties in different spheres of life, allowing the tensions that these spheres generated to be articulated, is largely absent from the school environment. In fact, the school trajectories of several of the people we interviewed—rather short, finishing before finishing primary school—showed the little hope they had in the school environment. For them, school was a short waystation prior to their premature incorporation into the world of work, in the case of men, or to agricultural and domestic tasks in that of women. "I, unfortunately", said one of them, "in my house of 10 people, the most educated was me, who was in the 4th grade of primary school (I had not therefore finished primary school, which was made up of 6 grades). At that time there were other values" (LF: *retired small businessman. Four children. Primary education. Poor social origin*). "Most of my colleagues", commented another, "who had the same opportunities that we had, had parents who did not know how to read and who did not know how to value the possibilities that the school had. So, I did the third class when I was 9 years old" (LF: *woman with six children. Primary education not completed. Suitcase factory employee. Widow. Very poor social origin*). "There was no obligation to study, and I didn't study", said another of our informants, "and then my mother made me learn sewing" (LF: *woman with two children. Primary education, retired typist secretary from a driving school*). "Those who finished the fourth year (primary education without completing)", another told us, "left their studies, because they were already tired of school and wanted to go to work. And I also got to work early. The value of studies had never been instilled in us" (FG: *female primary education. Two children. School janitor Retired. Very poor social origin*). And in this same direction, another of the people interviewed said that "before April 25 (the date of the Portuguese Carnation Revolution, which started democracy and the modernization of the country) the majority of parents said that their children, starting from the 4th grade (unfinished primary school), had to go to work to help raise the other siblings, because if they continued studying they lost the possibility of helping the family" (FG: *woman, retired textile industry worker, with primary studies. Low origin class*). "In my house of 10 people, the most cultured was me, who was in the 4th grade of primary school. At that time there were other values" (LS: *retired small businessman. Four children. Primary education. Poor social origin*). And among these values, as we said, the school value did not have a main place, especially for the children of families with the humblest social status, who did not see the possibility of improving their social position through the meritocratic school route. In fact, in the survey we carried out, when the people surveyed were asked to indicate the three aspects that they valued most of those that were transmitted to them at school, one of the least valued answers was the "*importance of a title to work*" (*see survey, graph 4*).

The school universe, disconnected from the fulfilment of social and moral duties and impregnated with a religious sense that gave meaning to other spheres of their lives, therefore provided little hope to the people we interviewed, especially to those of humbler social status, but also created few frustrations. "They did not expect anything they had not obtained", and thus, "they did not obtain anything they had not expected" (Bourdieu and Passeron 2001, p. 223). Devoid of those moral references, of those horizons of significance (Taylor 1996, pp. 43 ff.) linked to the values of *honesty* and *trustworthiness*, *respect*, *solidarity*, and *justice*, little good was remembered from school, other than moments of liberation in the company of peers. "At school", said one person, "I still have the memory of some classmates, that we greet when we see each other, but nothing more than that" (LF: *woman with two children. Primary education, retired typist secretary at a driving school. Humble social origin*). Friendship with peers is precisely one of the aspects that is most valued by the

people of this generation that we surveyed, regardless of gender and their level of education (*see survey, graph 4*) (sex: $\chi^2$ (10) = 25,24, ns; education: $\chi^2$ (20) = 23,29, ns).

Furthermore, school was remembered by these people as a disciplinary, rigid, and authoritarian setting that, above all, instilled fear. "Inside the classroom", said one of the people interviewed, "there was a lot of strictness (...). At that time, they conveyed fear... I felt humiliated (LF: *female primary education. Two children. Retired school janitor. Very poor social origin*). "At that time", commented another, "there was respect for the teachers, even fear, because they were rigorous and severe (LF: *woman with two children. Primary studies, retired typist secretary from a driving school. Humble social origin*).

But even those of our interviewees from slightly more well-off families, who harboured expectations of social mobility that they finally ended up achieving, barely mentioned when referring to their school years any of the values mentioned above other than those referring to their desire to improve socially. "In primary school", said one of these people, "nothing special was transmitted to me. The most important thing I learned was perhaps the importance of work" (LF: *retired primary school teacher. Two children*). "There wasn't much proximity between teachers and students", said another person. "What there was, was a lot of discipline; Everything was absolutely planned, and whoever did not do what was ordered was punished (...) I told myself, I have to be better than my parents" (LF: *woman, retired primary school teacher, two children. Social origin of small agricultural owners*). "My father", commented another person, "wanted me to study so that I could have a better future, and to a certain extent I did" (LF: *man with two daughters, technical engineer; retired secondary school teacher. Parents who own farms with some workers*). The meritocratic value of social mobility was, therefore, what gave meaning to the school universe, making the disciplinary and authoritarian environment that reigned there more bearable. But this value was, above all, instrumental, a means to an end. Well, what gave coherence to the lives of these people, as they themselves told us, was the fulfilment of the values of *honesty*, *trustworthiness*, *respect*, *solidarity*, and *justice*, which, with a religious background, were fundamentally linked to the family, community, and parish worlds and to the work world and which they also tried to transmit to their children. "In society", said one of these people, "we have the obligation to defend a name, to respect and to be respected (...) My daughters retain many of the values that we have transmitted to them, especially because they are honest people, they are ashamed; those things that are not judged now" (LF: *man with two daughters, technical engineer; retired secondary school teacher. Agricultural owner parents with some workers in their care*). And another person commented, "the values that I have transmitted to my children, associated with religious practice, are there; the values of honesty, respect..." (LF: *woman, retired primary school teacher, two children. Social origin small agricultural owners*). Some of these values, particularly those of responsibility, honesty, and trustworthiness, were also quite appreciated by the people of this generation that we surveyed (*see survey, graph 2*).

Moments of leisure were remembered by the people we interviewed, during the short period of their childhood, as a moment of liberation from the harshness of life. "I remember freedom", one of these people told us. "With freedom we were like birds (...). We were happy, we didn't feel tied down" (LF: *woman with primary education. Two children. School janitor. Retired. Very poor social origin*). "Half a dozen kids would get together", said another, "we would walk to the river to bathe; there was misery, but we were happy in life (...). It was being with each other, forgetting the discomfort of an empty stomach" (LF: *retired typographer, primary education, three children, poor social origin*). But when this last person was asked about the reasons for his happiness, in addition to the feeling of liberation from the sorrows of life that leisure provided, the values that gave, and still give, meaning and stability to his world came to light. "There was that, how can I say, there was peace and there was respect, which existed at that time, not so much today. And we had fun with each other" (LF: *retired typographer, primary education. Three children, poor social origin*). Already in youth, those moments of liberation from childhood were reduced to the rare moments in which one could forget a little about the fatigue produced by daily work and miseries. "I

left at 7 in the afternoon, then I had to make food for the next day, and it was not possible. But on the weekend, I would go for a little trip", said one of our informants (LF: *woman with two children. Primary studies, retired typist secretary from a driving school. Humble social origin*). And another said, "we went to eat the chicken and the roast on Sundays. And I remember a lot of those good things that I had in life" (LF: *woman with six children. Primary education not completed. Suitcase factory employee. Widow. Very poor social origin*). Another man commented, "we went to parties; At that time in those parishes around the city there were parties, various parties, and there was a man who played music to dance to, and we asked him for a dedicated record. It was what there was. This way we would forget about the discomfort in our stomach (he said it with a broad smile). That was the way to have fun" (LF: *retired typographer, primary education. Three children, poor social origin*).

We have shown so far how, for the people of the 1940–1950 generation that we interviewed belonging to the geographical area of *Braga* in *Portugal*, that religion, in which practice and belief are closely linked, provides meaning to the values that organise and give meaning to the relationships that these people maintain with their natural environment and with other people in the different spheres in which their daily lives unfold. We have also shown how religious beliefs allow these people to face the tensions that arise in their lives. Through the relationship they establish between the sphere of the transcendent and the immanent of their religious life and their ordinary life, they create bridges of stability, trust, and security in the midst of the oceans of precariousness and insecurity in those in which they have had to live (Arendt 1999, p. 106).

The way that the people we interviewed have of relating to their natural and human environment, of understanding it and of judging it, through their religious beliefs, which gives meaning to their main moral and social values, is the one that their children no longer have, for whom the received belief is no longer in relation to their practice. Their way of understanding and judging the world thus becomes more reflective with respect to received religious beliefs and with less capacity to reconfigure their own experiences and life situations. (See Figures 1–7).

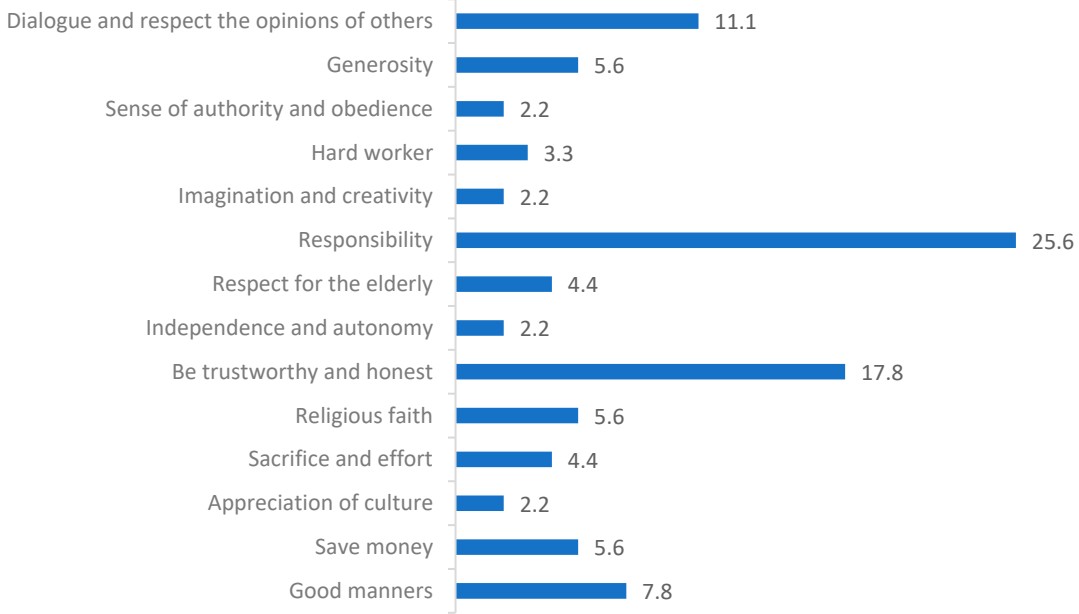

Basis: Individuals born in the 1940s and 1950s.

**Figure 1.** Name the three most important qualities that your family passed on to you.

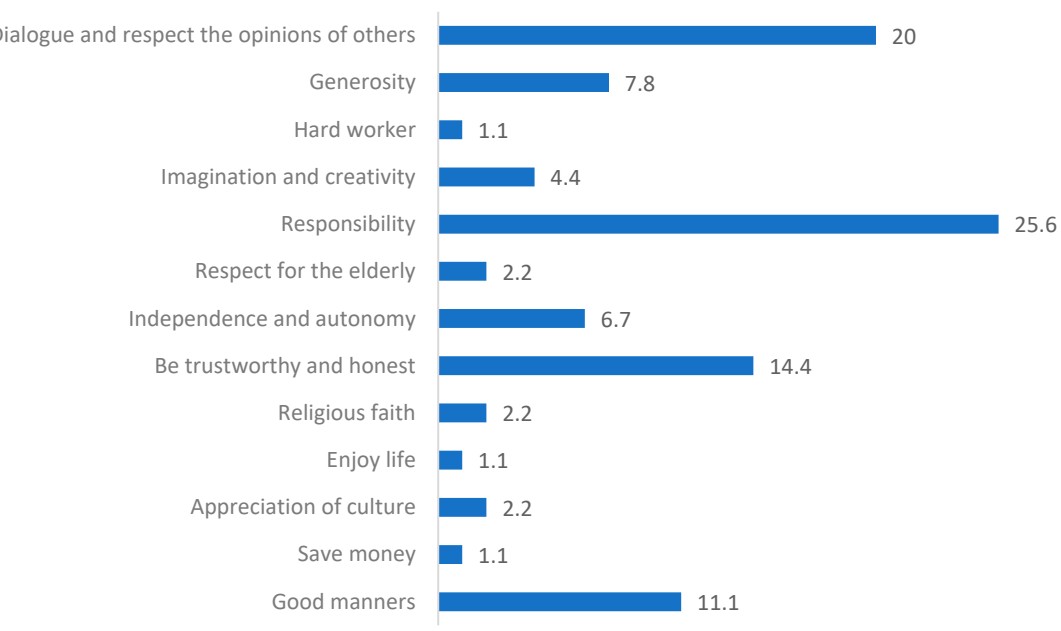

**Basis**: Individuals born in the 1940s and 1950s.

**Figure 2.** Of these qualities, indicate the three that you consider most important to instil in your children.

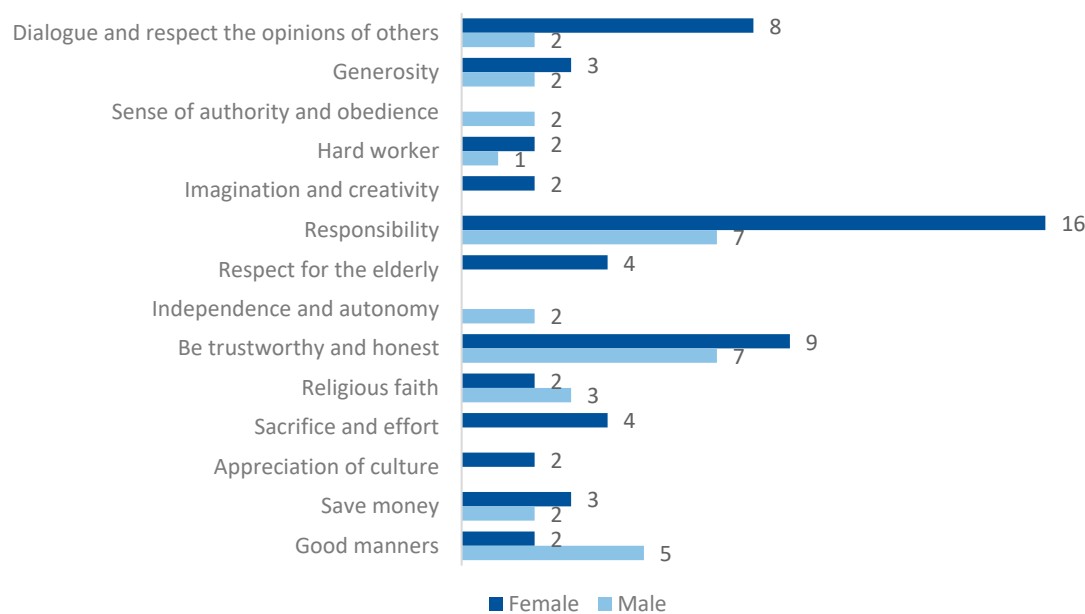

**Basis**: Individuals born in the 1940s and 1950s.

**Figure 3.** Name the three most important qualities that your family passed on to you, depending on your gender.

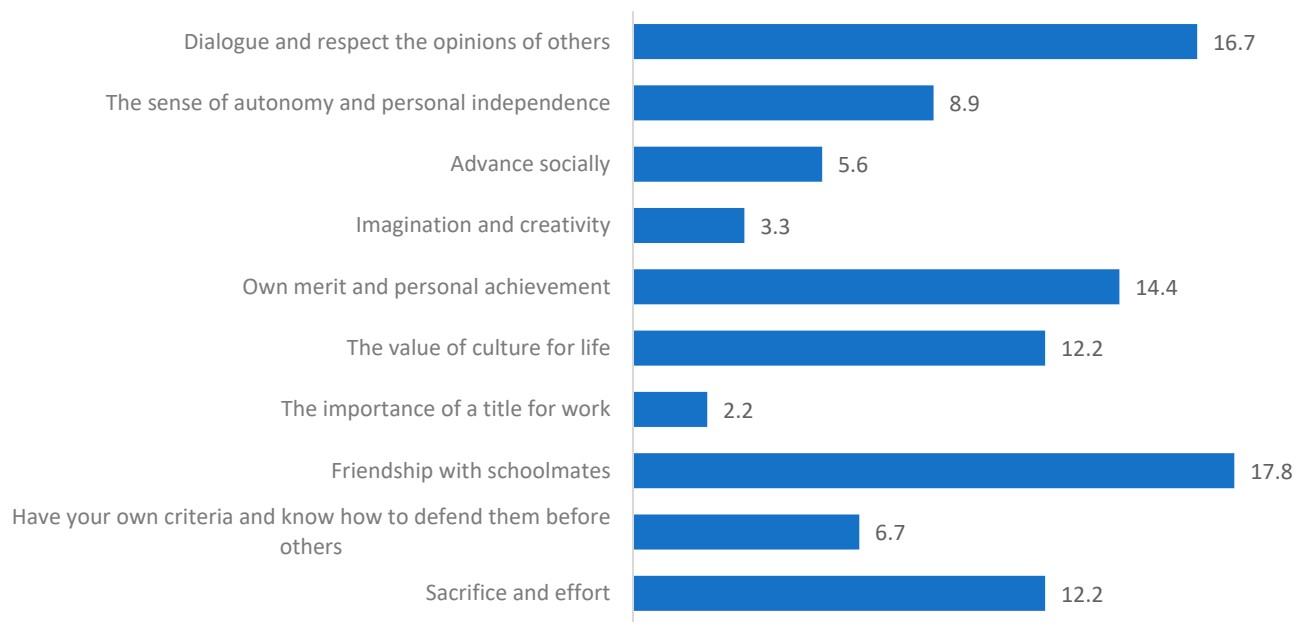

**Basis**: Individuals born in the 1940s and 1950s.

**Figure 4.** Of the aspects that people consider important in life, indicate three that you value most and that were passed on to you at school.

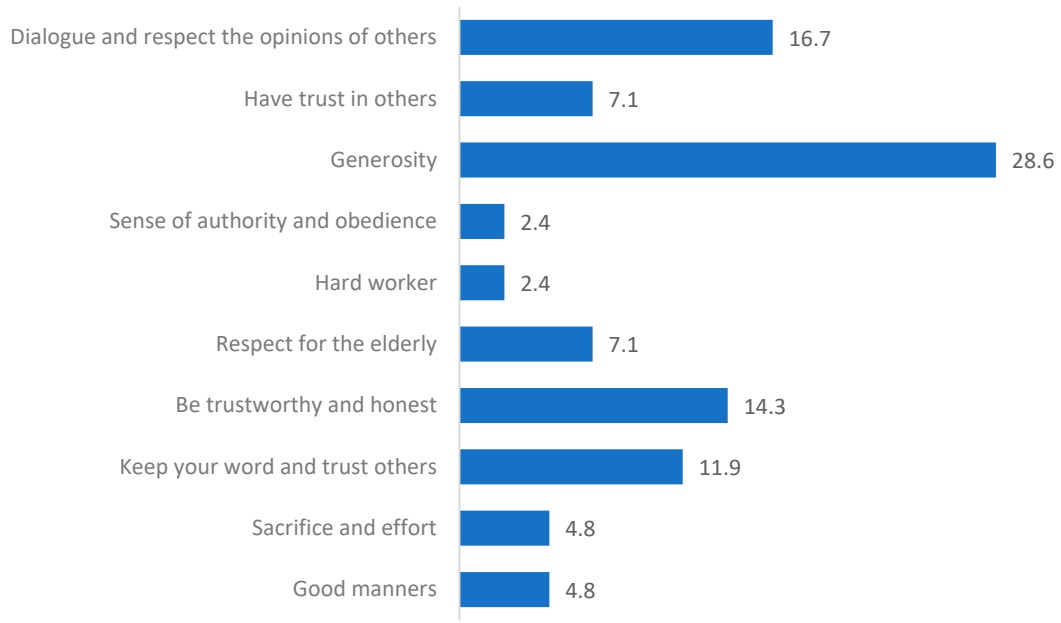

**Basis**: Individuals born in the 1940s and 1950s.

**Figure 5.** Indicate the three most important values you received through religion.

Name three qualities you valued most in your professional life.

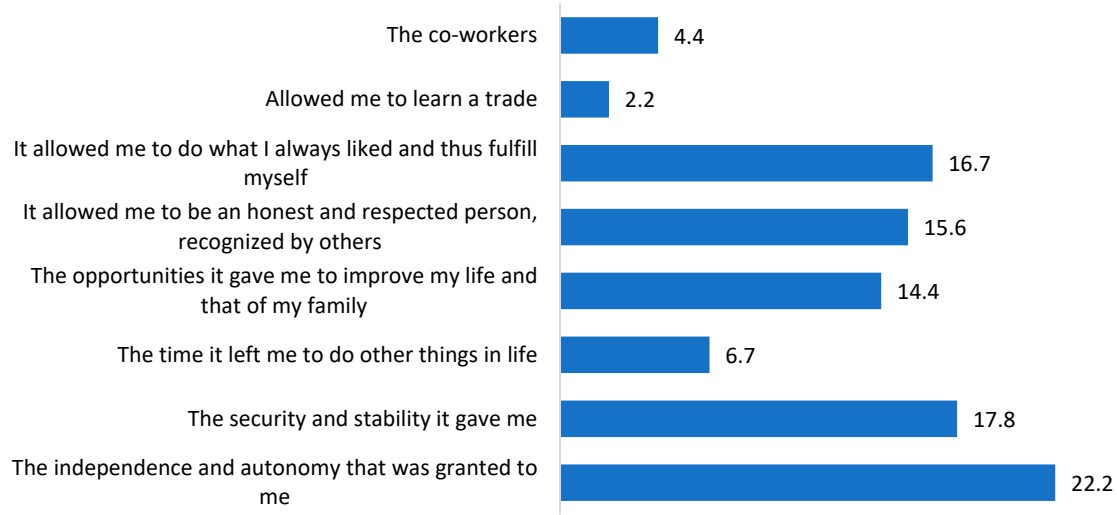

**Basis**: Individuals born in the 1940s and 1950s.

**Figure 6.** Name three qualities you valued most in your professional life.

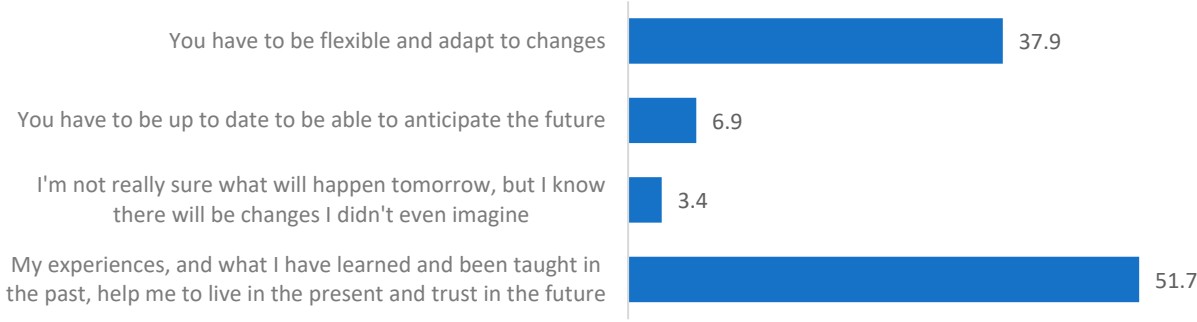

**Basis**: Individuals born in the 1940s and 1950s.

**Figure 7.** Indicate which of the following phrases you agree with most.

## 5. Discussion

The results obtained in our research on people born in the decade of *1940–1950* in the geographical region of *Braga* in *Portugal*, described above, confirm several of the hypotheses suggested in said research.

The first of them is related to the *concept of generation*. Although the people we investigated were structured based on gender and social class, in addition to their generational affiliation, this last dimension has been revealed as the most relevant. Indeed, although social class differences have been perceived regarding expectations of social mobility, the main values that articulate and give meaning to the vision of the world of the people of this generation were quite common to all of them, regardless of class and gender differences. These values were those of *honesty* and *trustworthiness*, *respect*, *solidarity*, *justice*, *responsibility*, and *work*, all of them, as has been seen, impregnated with a religious sense. These values have been acquired and internalised by the members of this generation through a series of common experiences and life situations, creating a certain generational consciousness. This consciousness has been formed from what was received by previous generations in close contact with what was experienced by these people during their childhood and youth, which reconfigured subsequent experiences and life situations (Mannheim 1993).

Now, as has been revealed in the approach of our research, this common generational consciousness is not presented as a will to transform the world in relation to a certain

social or political project (Aboim and Vasconcelos 2014). It manifests itself, above all, as a series of practices and representations that, as Bourdieu said, do not involve the conscious pursuit of certain ends (Bourdieu 2007, p. 86), but they carry a series of convictions about their respective life projects about what has and does not have value, what should and should not be done, and what is or is not worth it. From this point of view, the people of the generation we investigated have a common identity, which is expressed in the way they understand their lives through how they narrate it in their *life stories*, showing how they have become what they are through certain efforts and sacrifices, achievements, and failures (Taylor 1996, pp. 64–65).

Through these narratives, the people of this generation showed, as suggested at the beginning of this article, that the ways in which they understand their biographical trajectories, throughout which they have maintained a certain relationship with other people and with its natural environment, is influenced by religion, in relation to the main values that give meaning to these life trajectories. For these people, religion is an experience in which belief and practice are closely linked, and this is how they also received it from their parents. They therefore participate in a popular religiosity (Taylor 2014, vol. I, p. 112), which makes sense through the different rituals that make up the religious experience (Rappaport 2001; Douglas 1988). Their universe is that of belief. They are therefore completely far from any form of secularism.

Through their religious beliefs and the values associated with them, these people conceive themselves as part of their world and not as individuals facing the world. They do not have a differentiated self but one fused with their own world (Bellah 2017, p. 72). Their identity is completely linked to the universe they inhabit. By virtue of all this, the relationships they maintain with other people and with the natural environment in which they live are, above all, ones of interdependence. They are not conceived as autonomous and self-sufficient people, called to express their transformative and creative capacity; they are completely far from a supposed ethics of authenticity (Taylor 1994). On the contrary, they conceive themselves in debt to other people and to their natural environment, without which they do not believe they could have become what they have been in life. They thus manifest a morality of care and help, presided over by the values of *respect*, *honesty* and *trustworthiness*, *solidarity*, *justice*, and *responsibility*, a morality that is also hierarchical, since it does not understand the relationship with the world or with other people but through the connection with a previously established order that must be respected. Raised in traditional agricultural communities, they have a special bond with the land, over which they do not exercise any will to dominate but, rather, accept a liberating dependence. The earth is understood, in effect, as an almost sacred reality offered by the Creator for sustenance, and agricultural work is carried out, therefore, with an attitude of gratitude and admiration, with the awareness that the fruits that the earth provides are due, above all, to its potential fertility, to which human beings can only contribute with their work and effort. This way of understanding the relationship with the land is, therefore, still premodern. It is linked to a long tradition that goes back to the Greco-Latin and Christian world (Arendt 1998; Vernant 1985; Le Goff 1983), a tradition with which the modern mentality breaks, particularly since John Locke (Locke 2006, pp. 67–69), for whom human work becomes the main cause of the productivity of the earth, starting a process at the end of which will be human work, with its capacity to transform nature for its own benefit, which results in sacralisation. This productivist attitude is, as has been shown, completely contrary to that of our informants, who understand the relationship with the earth from a point of view closer to that of *eco-theology*.

The perspective of interdependence and respect for the land and the community is deeply rooted in the religious beliefs and practices of those interviewed in this study. This community *ethos* is echoed in the vision of Thomas Merton (1948), a Christian theologian and Trappist monk, who wrote "*No Man Is an Island*". The concept suggests that our identity is not just a product of our individuality but is also intrinsically linked to the community and land we inhabit.

The Liberation Theology, particularly Leonardo Boff (2015), developed an entire theory about universal fraternity and preferential care for the oppressed and for the land, stating that the land should not be seen just as an object to be explored but as the mother who welcomes us and who needs to be cared for and loved in return. This theological understanding serves as a bridge between spiritual life and social practice, merging immanence and transcendence in a unique and solid way.

Religion and religious beliefs to which the main values that give meaning to the lives of these people are linked (those of *respect, honesty* and *trustworthiness*, *solidarity*, *justice*, and *responsibility*) reinforce their confidence, in a context full of uncertainties in which a good part of their lives has passed. Through these beliefs, they create stability and order where only chaos would reign, a stability that also extends over time and presides over the ecclesiastical cycle and the harvest cycle, according to the tradition of received values and, also, to that related to the sphere of transcendence and the sacred, a superior time that orders and confers stability on the world (Taylor 2014, vol. I, p. 109). In this way, they create bridges of security that link the future, the present, and the past (Adam and Groves 2007, p. 47).

In this sense, religious beliefs not only guide the concrete actions of daily life but also provide a sense of stability and order in a world full of uncertainty. Just as religious experience points to an ethic of care and responsibility for the *common home* (no. 1 LS), our interviewees already live this ethic in their daily lives, guided by the values of respect, honesty, and justice. *respect, honesty, and justice The intersection of these tw*o universes, the theological and the practical, offers a richer and more nuanced understanding of how faith and everyday life can be intrinsically interconnected.

In the stories of their lives, the relationship they establish between the sphere of the transcendent and the immanent is very present, an ordering relationship, because it allows them to face the tensions that life generates in the different areas of their existence. Those other areas, such as school, in which this link almost disappears because the expectations placed in the sphere of the immanent are practically non-existent, mainly in people of humbler conditions, or because even though they exist, in the case of the children of more affluent families, are purely instrumental; these areas, we have said, have less capacity to create horizons of significance that guide people's lives in a certain direction, as they are not associated with the strong values mentioned above (Taylor 1996, p. 42).

By virtue of everything said up to this point, it can be stated that religion is, for the people we interviewed, in close connection to their system of values, to which it gives a special meaning, an effective protector against anomie. In fact, one of the people we interviewed told us that, during the pandemic, unable to continue going to church, as was her custom, she suffered such a personality crisis that she had to be admitted to a psychiatric hospital (LF: *woman with six children. Primary studies unfinished. Suitcase factory employee. Widow. Very poor social origin*). Without going to such extremes, religion has allowed these people to find meaning where life, in some of its phases, seemed to deny it time and again with its extreme harshness. Their firm belief enabled them to continue forward without too many doubts, knowing who they were and where they had to direct their lives, enduring all kinds of sacrifices (Berger 1977, p. 120). That confidence was what they also wanted to pass on to their children.

However, they have become more reflective, when confronting the legacy received with their own experiences and life moments in a world in which religious belief appears as one possibility among others, such as non-belief (Taylor 2014, vol. I, p. 23). According to what our informants told us, their children do not openly deny their parents' beliefs, but rather, they gradually distance themselves from them as they form less and less a part of their own life experiences. They have a certain attitude of estrangement towards them. This does not allow us to affirm that they no longer have any meaning for them; perhaps that meaning has been transformed and reconfigured without being completely lost. All of which requires a deep investigation into the life trajectory of the people from the following generations, a task to which we have oriented our present and future research with the

purpose of finding out what role the universes that we have analysed in this article play in these trajectories as creators of meaning, taking into account that human beings cannot live their lives if they appear to them as meaningless (Berger 1977, p. 127).

**Author Contributions:** The conception, design, methodology, writing and review were developed collaboratively by E.D. and J.F.D.V. All authors have read and agreed to the published version of the manuscript.

**Funding:** This research received no external funding.

**Institutional Review Board Statement:** Not applicable.

**Informed Consent Statement:** Not applicable.

**Data Availability Statement:** Data are contained within the article.

**Conflicts of Interest:** The authors declare no conflict of interest.

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
