# Peer review of "Generations and Life Worlds: The Case of Braga in Portugal"

_religions, doi:10.3390/rel14111413_

Round 1

Reviewer 1 Report

Comments and Suggestions for Authors

This is a thoroughly researched and thoughtfully presented paper, which gives insights into the lifeworlds of people born between 1940 and 1950 in the region of Braga, Portugal, part of a multi-generational study. It demonstrates the explanatory value of the concept of generation.

My main concern is with the balance of the piece, whether the empirical material is presented in sufficient depth and breadth to justify the well-theorized conclusions drawn from it. To take a single example, gratitude towards nature, the authors argue that 'the relationship with the land is dominated by a certain attitude of gratitude' (512-3), and this is taken as  pretext for nearly 2 pages of exposition which contrasts broadly harmonious pre-modern attitudes to nature drawing on classical sources and the Benedictine rule with modern agonistic approaches traced to Locke, which is in turn contrasted with recent papal eco-theological statements, which are  thus presented as both rooted in a long pre-modern tradition and resonating the discourse of the researched generation.

Yet the only trace of gratitude I could find in the reported speech of respondents is in a single passage attributed to 'LF, retired small businessman' (509), who says in relation to the plants in the small garden he tended in his retirement, 'They begin to grow (...) I begin to think, my interior and my mind ended up being more humanized. I treat that with affection.' From this, I think an attitude of gratitude may reasonably be inferred, but the context is one of a small domestic garden as a retirement project and is not obviously linked to the childhood experience of subsistence farming, let alone to sense of being part of some medieval-style vision of the Great Chain of Being. 

I am not saying that the authors interpretation of respondents testimony is wrong - they have access to all the material and context and I only have what they have written - but it does not seem to me to be sufficiently justified by the evidence presented. 

So I would recommend rebalancing the article to present more primary evidence and less secondary analysis and commentary, especially for the sections on attitudes to nature and religion, if the article is to be considered a social scientific rather than a theological contribution.     

Comments on the Quality of English Language

In many ways the article is well written, indeed sophisticated, but needs checking by a native or fluent English speaker for errors e.g, in the abstract line 1 'a research on' should read either 'research on' (no article) or 'a research project on', and for unduly lengthy phrasing e.g.  line 1 of the introduction 'has been analysed early on, in the first decades of the last century' should be changed to 'was analysed in the early twentieth century' or similar. 

Reviewer 2 Report

Comments and Suggestions for Authors

1) It would be useful to note the particularity of Catholicism (Spain-Portugal) in relation to Protestantism with regard to the meaning attributed to tradition in the generation of the 40s and 50s, as opposed, for example, to Anglo-Saxon individualism (utilitarian/expressive individualism: Bellah).

2) Mention should be made of the role of the spirit of personalized self-reflexivity (Giddens, Beck) in the dismantling of the meaning of the values of tradition for the generation of the 40s and 50s.

3) It would be appropriate to allude to the responsibility (and crisis) of "intermediate institutions", as mediators between commonly established meanings and individuals, in anomic phenomena. See: "Modernidad, pluralismo y crisis de sentido", Barcelona, Paidós, 1997.

4) In the body of the text, sentences in Spanish appear interspersed. This should be corrected.

Reviewer 3 Report

Comments and Suggestions for Authors

Dear Author / Authors,
Your work is very interesting.
It undoubtedly fits into the trend of research on values. In my review, I only want to write about what I think needs to be supplemented or changed.
1. I did not find in the initial part of the work the aims of the article. Only in line 837 there is a reference to previous hypotheses. Unfortunately, I couldn't find them in the text earlier.
2. The article is missing conclusions.
3. The article analyzes religious values. Unfortunately, it is not known what religion these values refer to, because there is no such information in the text of the article. You can guess that this is also the Roman Catholic Church.
4.
I think it would be better to sort / arrange the results in the charts by value (from the largest to the smallest).
5.
I have marked other minor comments in the file.

Round 2

Reviewer 3 Report

Comments and Suggestions for Authors

I have no comments.